# Hospital-wide natural language processing summarising the health data of 1 million patients

Daniel M. Bean[1,2]*, Zeljko Kraljevic[1,3], Anthony Shek[1,4], James Teo[4,5], Richard J. B. Dobson[1,2,3,6,7]

1 Department of Biostatistics and Health Informatics, Institute of Psychiatry, Psychology and Neuroscience, King's College London, London, United Kingdom, 2 Health Data Research UK London, University College London, London, United Kingdom, 3 NIHR Biomedical Research Centre at South London and Maudsley NHS Foundation Trust and King's College London, London, United Kingdom, 4 Department of Clinical Neuroscience, Institute of Psychiatry, Psychology and Neuroscience, King's College London, London, United Kingdom, 5 Department of Neuroscience, King's College Hospital NHS Foundation Trust, London, United Kingdom, 6 Institute for Health Informatics, University College London, London, United Kingdom, 7 NIHR Biomedical Research Centre, University College London Hospitals NHS Foundation Trust, London, United Kingdom

* daniel.bean@kcl.ac.uk

**Data Availability Statement:** The full patient-level dataset contains sensitive and potentially re-identifiable data and cannot currently be made available directly. To support data availability, we

## Abstract

Electronic health records (EHRs) represent a major repository of real world clinical trajectories, interventions and outcomes. While modern enterprise EHR's try to capture data in structured standardised formats, a significant bulk of the available information captured in the EHR is still recorded only in unstructured text format and can only be transformed into structured codes by manual processes. Recently, Natural Language Processing (NLP) algorithms have reached a level of performance suitable for large scale and accurate information extraction from clinical text. Here we describe the application of open-source named-entity-recognition and linkage (NER+L) methods (CogStack, MedCAT) to the entire text content of a large UK hospital trust (King's College Hospital, London). The resulting dataset contains 157M SNOMED concepts generated from 9.5M documents for 1.07M patients over a period of 9 years. We present a summary of prevalence and disease onset as well as a patient embedding that captures major comorbidity patterns at scale. NLP has the potential to transform the health data lifecycle, through large-scale automation of a traditionally manual task.

## Author summary

Clinical notes and letters are still the main way that medical information is recorded and shared between clinical staff. This means that for research we need methods that can cope with text data, which is typically far more challenging than "structured" data like diagnosis codes or test results. In this study we apply a state of the art clinical text processing model to analyse almost 10 years worth of text data from a large London hospital, covering over 1 million patients. We are able to find patterns of disease burden, onset and co-occurrence

have registered the dataset on the Health Data Research UK Innovation Gateway online (https://web.www.healthdatagateway.org/dataset/4e8d4fed-69d6-402c-bd0a-163c23d6b0ee). This provides a thorough description (with the highest tier metadata score awarded by the platform(35)) and procedure for access of summary or aggregated data (to fully eliminate re-identification risk).

**Funding:** The project has received funding support from Innovate UK, NHS AI Lab, Office of Life Sciences, Health Data Research UK, NIHR Maudsley Biomedical Research Centre and NIHR Applied Research Centre South London. DMB is funded by Health Data Research UK and NHS AI Lab. RJBD is supported by the following: (1) NIHR Biomedical Research Centre at South London and Maudsley NHS Foundation Trust and King's College London, London, UK; (2) Health Data Research UK, which is funded by the UK Medical Research Council, Engineering and Physical Sciences Research Council, Economic and Social Research Council, Department of Health and Social Care (England), Chief Scientist Office of the Scottish Government Health and Social Care Directorates, Health and Social Care Research and Development Division (Welsh Government), Public Health Agency (Northern Ireland), British Heart Foundation and Wellcome Trust; (3) The BigData@Heart Consortium, funded by the Innovative Medicines Initiative-2 Joint Undertaking under grant agreement No. 116074. This Joint Undertaking receives support from the European Union's Horizon 2020 research and innovation programme and EFPIA; it is chaired by DE Grobbee and SD Anker, partnering with 20 academic and industry partners and ESC; (4) the National Institute for Health Research University College London Hospitals Biomedical Research Centre; (5) the National Institute for Health Research (NIHR) Biomedical Research Centre at South London and Maudsley NHS Foundation Trust and King's College London; (6) the UK Research and Innovation London Medical Imaging & Artificial Intelligence Centre for Value Based Healthcare; (7) the National Institute for Health Research (NIHR) Applied Research Collaboration South London (NIHR ARC South London) at King's College Hospital NHS Foundation Trust; (8) NHS AI Lab. The funders had no role in study design, data collection and analysis, decision to publish, or preparation of the manuscript.

**Competing interests:** I have read the journal's policy and the authors of this manuscript have the following competing interests: JTT has previously received research grant support from Innovate UK,

purely in text data. This result strongly supports the use of clinical text data in research and provides a summary of the scale and nature of clinical text to other researchers.

## Introduction

Electronic Health Records (EHRs) are now widely deployed, and in many cases these electronic systems have accumulated a considerable history of clinical data. Each clinical site therefore represents a potentially significant data resource. There is considerable structured coding of clinical events and related results, and the structured data capture is highly targeted to specific purposes (primarily billing or reporting). Such structured diagnosis lists, problem lists or test lists often only partially capture the full clinical picture of a patient as the primary means of clinical communication and documentation is in the form of free text letters, notes and reports [1]. Most analytical quantitative research have focused on the structured elements only as the unstructured free text recorded in EHRs have traditionally been difficult to access and analyse [2–4]

In conventional healthcare workflows, both structured and unstructured aspects of EHR's are read by business intelligence staff and translated into standardised codes (termed 'clinical coders') for submissions into datasets. Structured data can be analysed at a regional or national level to gain powerful insights into clinical trajectories at scale [5,6]. This largely manual process uses the ICD10, OPCS ontologies and follows rules around conciseness. Due to the laborious nature of this process and lack of an automation-assisted process, most organisations only perform this 'structuring and standardising' process on inpatient episodes and the text generated from the large proportion of outpatient activity are ignored. This 'lacune' means that certain populations with conditions that do not result in hospitalisations (or where clinical pathway transformations migrate to ambulatory or outpatients routes) would be under-represented systematically; dependency on only manually-derived coded data potentially incorporates a hidden 'inclusion bias' in many datasets.

Natural Language Processing (NLP) combined with rich clinical terminologies such as SNOMED have the potential to automate a large portion of the 'structure and standardise' process to make the full clinical record accessible to computational analysis [7–9]. Previous attempts have focused on specific cohorts (e.g. critical care patients only [10], patients with a certain disease only [11–13], discharge letters only [14]). Doing this across a whole hospital's record has not previously been attempted, and produces the opportunity to automate a laborious manual process for healthcare delivery, and also to enrich any structured registries or databases (like HES [15], SUS [16], CPRD [17], Caliber [18], CVD-Covid-UK [19]) with greater phenotypic and narrative expressiveness. Any downstream data-dependent activity, including population health and research, or trial recruitment [20], would potentially benefit.

In this paper we present the first descriptive summary of the entire text record of a large UK secondary and tertiary healthcare system in London, King's College Hospital NHS Foundation Trust over a period of about 9 years. To our knowledge this is the first study of a large-scale EHR dataset derived from NLP, although there are several other descriptive analyses of large-scale structured EHR data (Kuan et al. 2019; Thygesen et al. 2022; Kuan et al. 2023). Compared to structured data, the free-text portion of the EHR captures a more detailed clinical narrative. The description of this data provides three useful resources:

1. Detailed description of the scale and nature of the available data within a UK hospital

2. Analysis of disease prevalence and comorbidity patterns with comparison to national prevalence data in the NHS Quality and Outcomes Framework

NHSX, Office of Life Sciences, NIHR, Health Data Research UK, Bristol-Meyers-Squibb and Pfizer; has received honorarium from Bayer, Bristol-Meyers-Squibb and Goldman Sachs; holds stock in Amazon, Alphabet, Nvidia; and receives royalties from Wiley-Blackwell Publishing. DMB has received research funding from Pfizer. RJBD, ZK, AS declare that no competing interests exist.

3. Description of the data in the UK Health Data Research Innovation Gateway [21] to support open research and collaboration.

## Results

### Scale of data

We extracted data from 2011-01-01 to 2019-12-31 for all patients aged 18–100 years at the time of admission. The dataset includes 1.07M patients and 212M separate text notes. 157M NLP annotations to SNOMED concepts were generated for those notes. We found that the scale of data available has increased over time, partially due to an increase in the number of patients per year but, importantly, we also captured more data per patient over time. The number of patients in the dataset increased from ~165 to ~369k patients per year while the median annotations per patient per year increased from a median of 14 to 25. The increase of patient numbers is due to the merger of KCH with a suburban secondary care hospital (Princess Royal University Hospital, PRUH) in 2015, with subsequent data incorporation.

Fig 1 shows the breakdown of annotations by semantic type and meta annotation class. Most annotations were to SNOMED disorder, finding, substance and procedure classes, i.e. diseases, symptoms and treatments. Interactive treemap visualisations for the top 100 most common SNOMED codes detected for semantic types Finding, Disorder and Substance are available online [22]. The vast majority of annotations were positive (85.9%), referred to the patient (93.1%) and are current/recent in time (92.9%). We therefore focus on this subset of 123M annotations (78.3%) where all three criteria are true in the remaining analysis.

### Demographics

Data was extracted for 1.07M patients, 54.7% female with a mean age at first included admission of 46.95 ± 18.99 years. There was a small but significant age difference between male and female patients (n = 485603 males 47.69 sd 18.70 years; n = 587580 females 46.38 sd 19.22 years, t-test $p < 0.001$).

1904 SNOMED codes (11.8% of codes detected) had a significant difference in prevalence between male and female patients (t-test $p < 0.05$ after correction for multiple comparisons). Table 1 shows the top 10 most prevalent disorders overall and by gender. Fig 2 shows the prevalence of all 14 unique disorders from Table 1 across all groups. All differences in prevalence in Fig 2 and Table 1 were significant at $p < 0.001$ level. Many of the other differences in prevalence are expected sex-specific conditions.

We found the overall most prevalent concept was hypertension (14.65%) which was also the most prevalent disorder for both male and female patients. Note that these prevalence estimates are for single SNOMED codes (see "Comparison to National Data" prevalence estimates using code sets). However, the estimates are consistent with the QoF data for 2018–19 which found a prevalence of 13.96% for hypertension in England. Similarly, we estimate prevalence of asthma at 5.69% and the QoF estimate is 6.05% for England. However there are discrepancies, such as for depression (5.55% in our data, 10.74% in QoF for England, 7.65% for QoF in London).

### Comparison to National Data—NHS Quality and Outcomes Framework

We used the NHS England business rules (which define specific conditions using SNOMED codes [23]) and the NHS England Quality and Outcomes Framework (QoF) data to calculate prevalence of disorders included in the 2018/19 QoF report [24] according to the same

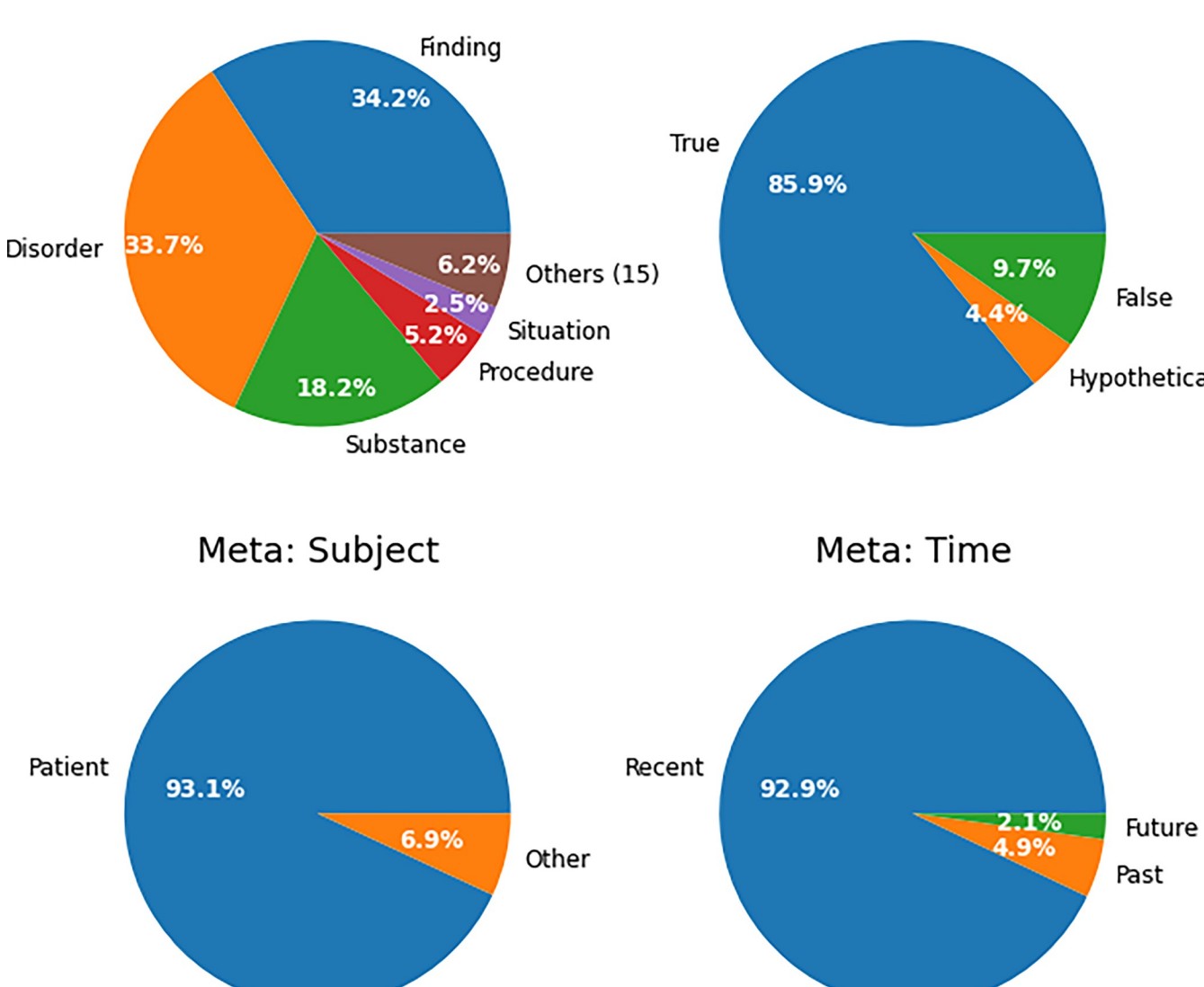

**Fig 1. Breakdown of semantic type and meta annotations for all entities.** Due to the large number of semantic types, only the top 5 are shown and the remaining 15 were combined into the "Others (15)" group for readability. Percentages are relative to all 157M NLP annotations.

definitions (i.e. sets of included codes) for each condition. The prevalence estimates for our data and the QoF results for England and London (for national and regional context) are shown in Table 2.

Across all conditions for which we could implement the same definition as used for QoF in our data we found a significant (p<0.001) difference in prevalence compared to primary care estimates for England and London. There were only 2 conditions for which we found a lower prevalence than the London estimate; CKD and Depression. In particular we find a rate of CKD that is less than half the London GP prevalence. 3782 patients in our data meet the QoF definition (CKD grade 3a to 5), and an additional 2546 patients for which we detected CKD but not a grade, so they cannot be included but could potentially have CKD grade 3a-5. Mild CKD may not have been explicitly typed by clinicians into letters or notes, but this under-

**Table 1. Top 10 most prevalent disorders overall and by sex.** Prevalence is shown in brackets. All prevalence comparisons between Male and Female groups were significant at p < 0.001 level with Fisher's exact test.

| Rank | All (n = 1073183) | Male (n = 485603) | Female (n = 587580) |
|---|---|---|---|
| 1 | Hypertensive disorder, systemic arterial (disorder) (14.65%) | Hypertensive disorder, systemic arterial (disorder) (15.44%) | Hypertensive disorder, systemic arterial (disorder) (14.00%) |
| 2 | Asthma (disorder) (5.69%) | Hypercholesterolemia (disorder) (5.92%) | Asthma (disorder) (6.44%) |
| 3 | Depressive disorder (disorder) (5.55%) | Contusion (disorder) (5.65%) | Depressive disorder (disorder) (5.90%) |
| 4 | Contusion (disorder) (5.43%) | Diabetes mellitus type 2 (disorder) (5.52%) | Contusion (disorder) (5.24%) |
| 5 | Hypercholesterolemia (disorder) (5.17%) | Depressive disorder (disorder) (5.12%) | Cyst (disorder) (4.99%) |
| 6 | Diabetes mellitus type 2 (disorder) (4.82%) | Cerebrovascular accident (disorder) (4.84%) | Urinary tract infectious disease (disorder) (4.65%) |
| 7 | Cerebrovascular accident (disorder) (4.34%) | Asthma (disorder) (4.78%) | Hypercholesterolemia (disorder) (4.54%) |
| 8 | Cyst (disorder) (4.27%) | Low blood pressure (disorder) (4.58%) | Osteoarthritis (disorder) (4.40%) |
| 9 | Low blood pressure (disorder) (4.25%) | Diabetes mellitus (disorder) (4.25%) | Diabetes mellitus type 2 (disorder) (4.24%) |
| 10 | Diabetes mellitus (disorder) (4.12%) | Cataract (disorder) (3.62%) | Arthritis (disorder) (4.24%) |

detection of CKD would be addressable using structured EHR data (i.e. eGFR blood result) as well as to map out the CKD to the different grades.

## Prevalence and onset of major diseases

In addition to the presence of a code, we can also analyse the age at first detection of a code in the record. Note that this will not always be age at diagnosis. 14 major conditions were manually identified to provide a breadth of clinical specialties and expected onset.

Fig 3 shows the age profile of the various diagnosis codes are compatible with expectations with diseases of young adults like Multiple Sclerosis (MS), Psychosis and Inflammatory Bowel Disease, while diseases of later ages being degenerative (dementia or Parkinson's Disease) or related to end-organ failure (heart failure). These age profiles suggest that large-scale NLP extraction from clinical documents produces datasets with similar characteristics to standardised national datasets.

Analysis of these diseases by sex finds many significant differences in both mean age at first detection and overall prevalence (Table 3). A particularly large, significant difference in first

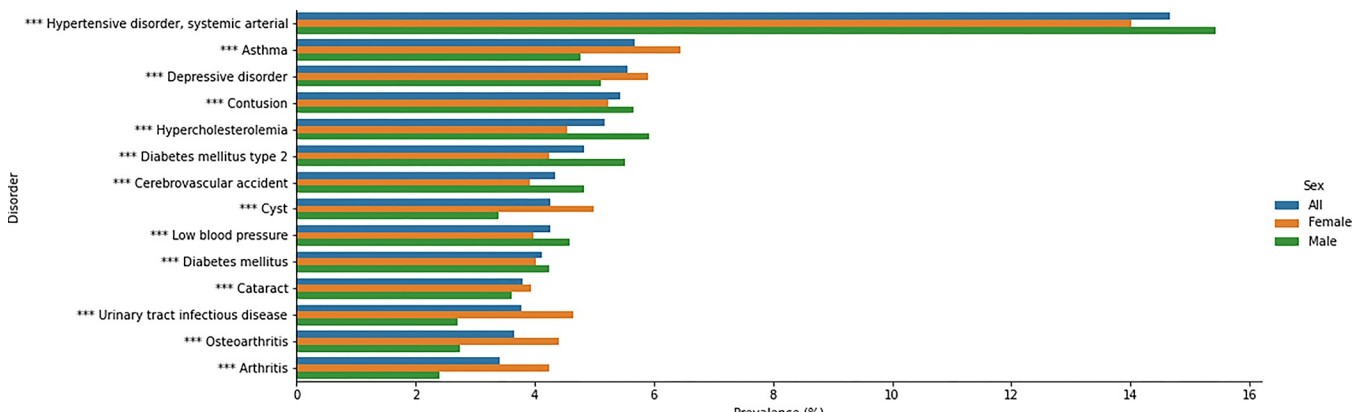

**Fig 2. Prevalence overall and by sex for the most prevalent disorders.** Prevalence is shown for all disorders that are in the top 10 most prevalent for any group. Colour indicates sex. All prevalence comparisons between Male and Female groups were significant at p < 0.001 level with Fisher's exact test. * = p < 0.1, ** = p < 0.01, *** = p < 0.001.

**Table 2. Prevalence estimates from our data and the UK 2018/19 Quality and Outcomes Framework data.** Prevalence estimates were calculated for all conditions for which the Quality and Outcomes Framework definitions could be directly mapped to our data including all admissions from 01 April 2018–31 March 2019. Conditions were defined using the NHS Digital business rules version 41 as used in the 2018/19 Quality and Outcomes Framework report. All pairwise comparisons between our data and the England or London estimates were significant at p<0.001 level following fisher's exact test and Bonferroni correction for multiple testing.

| Condition | Our Data Prevalence (%) | QoF England Prevalence (%) | QoF London Prevalence (%) |
|---|---|---|---|
| Asthma | 7.21 | 6.05 | 4.6 |
| Atrial fibrillation | 3.78 | 1.98 | 1.1 |
| CHD | 6.53 | 3.1 | 1.95 |
| CKD | 1.1 | 4.09 | 2.44 |
| COPD | 3.5 | 1.93 | 1.14 |
| Cancer | 9.56 | 2.98 | 2.03 |
| Dementia | 2.52 | 0.78 | 0.51 |
| Depression | 7.32 | 10.74 | 7.65 |
| Diabetes | 11.89 | 6.93 | 6.63 |
| Epilepsy | 2.88 | 0.79 | 0.55 |
| Heart failure | 1.9 | 0.93 | 0.56 |
| Hypertension | 19.34 | 13.96 | 10.96 |
| PAD | 0.87 | 0.6 | 0.33 |
| Rheumatoid arthritis | 1.66 | 0.76 | 0.55 |
| Stroke | 6.13 | 1.77 | 1.06 |

detection is noted for psychosis (46.12 for males, 50.87 for females, p<0.01). Although we do detect a younger first detection age for DM1 vs DM2 in both male and female patients, the age distribution for DM1 is not in line with expectations.

## Patient clustering captures high-dimensional comorbidity patterns

Each patient in the dataset can be represented as a vector of NLP annotation counts. These counts capture a complex interrelationship of comorbidity, treatment and outcome. To qualitatively capture these comorbidity patterns, we generated a low-dimensional embedding from the input vectors from a random sample of 100k patients and used agglomerative clustering with ward linkage to sample regions of this embedding space. The sample was not significantly different to the remaining patients in the dataset in their age (t-test p = 0.45) or sex (fisher exact test p = 0.90) distribution. As shown in Fig 4, the resulting embedding with 50 clusters contains regions of patients that are strongly associated with major diagnoses. For 72% of clusters, at least one SNOMED disorder was present in over 50% of cluster patients. For 48% of clusters, the most common disorder was present in at least 75% of the cluster patients. The annotations in Fig 4 include propagating counts through the SNOMED ontology (see Methods). An interactive visualisation of the embedding in Fig 4 is available online [22].

As we are only considering disorders, we are not attempting to uncover a globally optimal set of patient clusters. Instead we primarily use the clustering algorithm to sample contiguous regions of the patient embedding space which we can test for enrichment.

The colour assignment in Fig 4 should not be taken to mean categorical differences between two neighbouring clusters: for example we note two neighbouring clusters representing asthma patients with and without hypertension, and the cluster separation being an artefact of the clustering algorithm (e.g. patients in the asthma-hypertension group being simply on the same asthma cluster but on a spectrum). Although the clustering algorithm assigns hard borders between clusters, it is important that the embedding is a continuous space and we are not suggesting there are hard divisions between these patients.

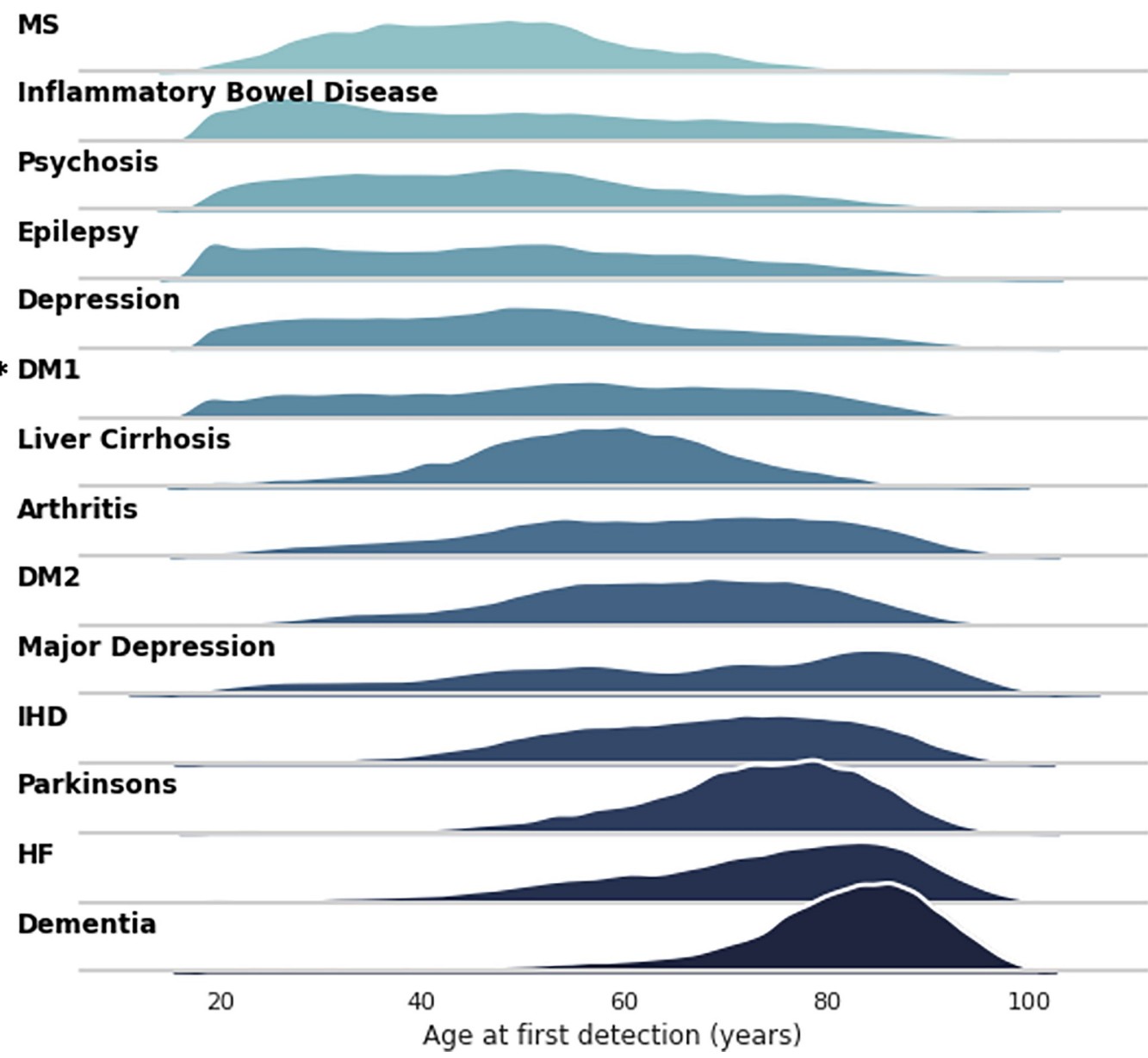

**Fig 3. Age at first detection for major diseases.** MS = multiple sclerosis, DM1 = diabetes mellitus type 1, DM2 = diabetes mellitus type 2, IHD = ischaemic heart disease, HF = heart failure. * = distribution not consistent with expectation. Colour indicates disease.

## Discussion

We present a summary of the text records for over 1M patients over almost a decade. We find that the rate of data capture per patient is increasing over time. The dataset is generated entirely by NLP applied to clinical text and captures major trends in disease prevalence and age of onset.

A number of significant differences in prevalence between male and female patients were observed. Depressive disorder was significantly more prevalent in female patients (5.90% vs 5.12%), as was asthma (6.44% vs 4.78%). In both cases the difference is consistent with expectations. We also find a significantly higher prevalence of dementia in female patients (2.47% vs 2.3%) and a later age at first detection (77.73 years vs 80.15 years). The greater prevalence of

**Table 3. Prevalence and age at first detection for selected major diseases.** Each disease is represented by a number of specific SNOMED codes. Ages are shown as mean with standard deviation in brackets. Prevalences are shown as percentages with counts in brackets. Age distributions were compared with a t-test and prevalences were compared with fisher's exact test.

| Disease | Mean age at first detection (years) | | | Prevalence [% (N)] | | |
|---|---|---|---|---|---|---|
| | Male | Female | p | Male | Female | p |
| MS | 47.04 (14.18) | 46.15 (13.99) | 1 | 0.24 (1155) | 0.44 (2606) | <0.01 |
| Inflammatory Bowel Disease | 49.40 (19.63) | 47.41 (20.44) | <0.01 | 2.99 (14540) | 3.06 (17983) | 0.65 |
| Psychosis | 46.12 (16.65) | 50.87 (18.65) | <0.01 | 1.14 (5530) | 0.71 (4152) | <0.01 |
| Epilepsy | 48.99 (18.97) | 48.06 (19.75) | <0.01 | 2.16 (10491) | 1.77 (10420) | <0.01 |
| Depression | 50.90 (17.85) | 49.72 (19.11) | <0.01 | 5.97 (28975) | 7.10 (41711) | <0.01 |
| DM1 | 55.29 (18.25) | 52.70 (20.17) | <0.01 | 1.49 (7250) | 1.19 (6984) | <0.01 |
| Liver Cirrhosis | 56.74 (12.85) | 56.41 (15.08) | 1 | 1.35 (6538) | 0.71 (4157) | <0.01 |
| Arthritis | 61.74 (17.31) | 63.34 (17.62) | <0.01 | 3.77 (18309) | 5.65 (33192) | <0.01 |
| DM2 | 64.71 (13.50) | 61.83 (17.47) | <0.01 | 5.57 (27047) | 4.28 (25123) | <0.01 |
| Major Depression | 64.53 (19.76) | 66.40 (20.84) | 1 | 0.13 (645) | 0.14 (837) | 1 |
| IHD | 66.55 (14.52) | 69.05 (15.92) | <0.01 | 6.06 (29448) | 3.29 (19357) | <0.01 |
| Parkinsons | 71.91 (11.81) | 72.37 (12.85) | 1 | 0.69 (3368) | 0.40 (2353) | <0.01 |
| HF | 71.00 (14.72) | 74.37 (15.62) | <0.01 | 1.68 (8148) | 1.12 (6572) | <0.01 |
| Dementia | 77.73 (13.28) | 80.15 (13.49) | <0.01 | 2.30 (11159) | 2.47 (14514) | <0.01 |

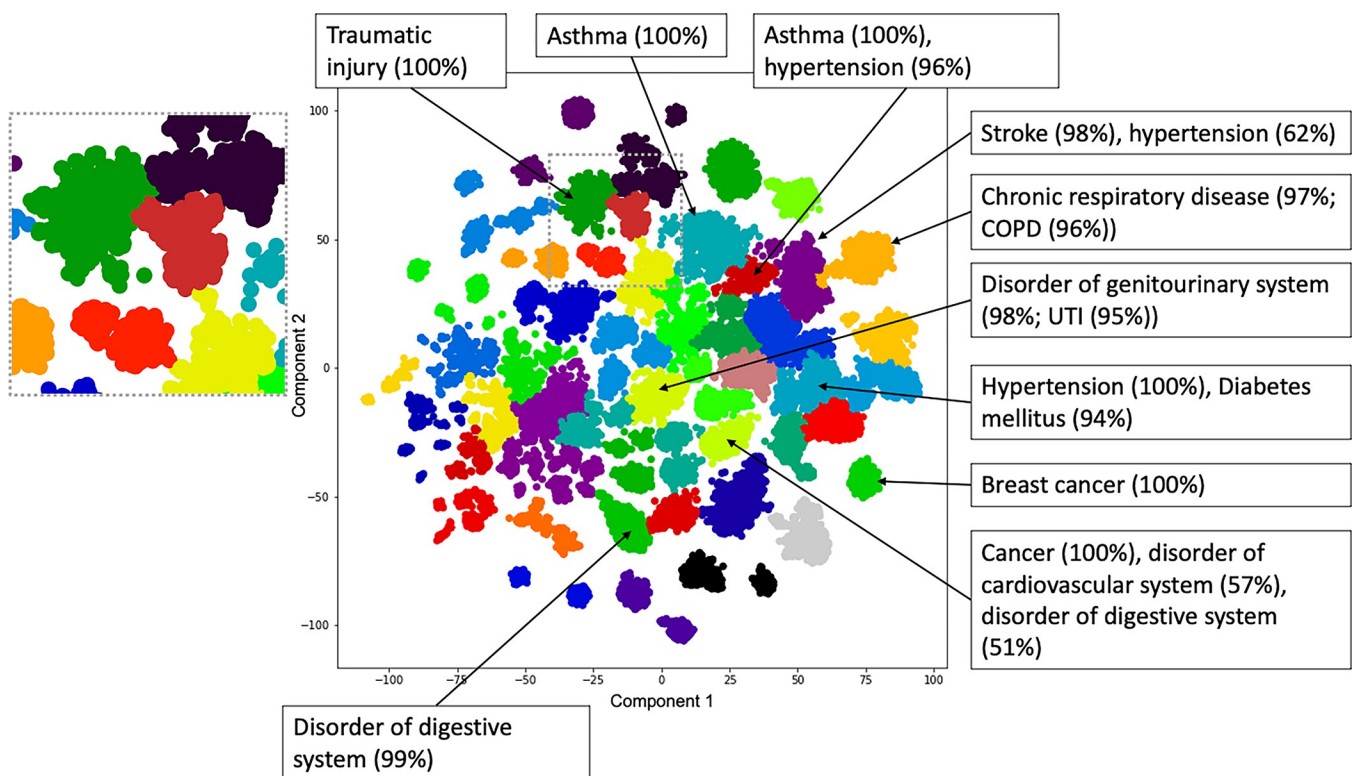

**Fig 4. Clustering of patients based on SNOMED disorder codes detected in free text.** A sample of 100,000 patients was embedded based on normalised annotation counts for all SNOMED disorder codes detected in at least 1000 patients at KCH. Colour indicates cluster membership (50 clusters), text boxes indicate major disorders in the indicated cluster with the percent of patients in brackets. Where a particular disease was predominant within a broader category it is also shown within brackets. The grey box on the left is a detailed view of the region indicated with a dashed grey border in the main plot area. COPD = chronic obstructive pulmonary disease, UTI = urinary tract infection. This visualisation is available in interactive form online [22].

dementia in female patients is well established. The age at first detection in our record is not necessarily the same as age of diagnosis, as a new patient could arrive with a known history of dementia. However, there is evidence that female dementia patients tend to be diagnosed later in life.

Prevalence estimates calculated for our data reflect a specific clinical context of secondary care inpatients. This cohort should be expected to have different prevalence of conditions from primary care (higher prevalence of most conditions) and also local differences in socio-demographic factors. Although we demonstrate many areas of strong agreement with expected trends from national data, there are discrepancies. A significant artefactual error is noted, eg: Type 1 Diabetes Mellitus which is known to have a young adult onset but this dataset captures a lot of middle-age and elderly onset cases. This is likely due to the NLP misattributing late onset Type II diabetes mellitus patients who are commencing on insulin—a vocabulary error of source document which is highly contextual to the specific snomed code as Type 1 and Type 2 due to the deprecated outdated concepts of insulin-dependent diabetes mellitus (IDDM) and non-insulin-dependent diabetes mellitus (NIDDM). This is correctable by setting the NER+L to recognise "IDDM" as an ambiguous token and not mapping it to "Type 1 Diabetes Mellitus (SCTID 46635009)". The high prevalence of cysts (rank 5 for female patients, 4.99%) is likely due to most mentions of cysts being annotated to the generic concept rather than a more specific term.

The strength of our dataset is its application to the in-hospital, secondary and tertiary care setting. We demonstrate that the large-scale analysis of NLP phenotypes can identify clusters of patients with differing major diseases, and find evidence that the embedding space captures clinical spectrums in some areas (e.g. patients with and without a major comorbidity). This suggests that outcomes of interest could be overlaid on the embedding space to find associations. The embedding is also potentially a powerful tool to identify data-driven cohorts of patients from high-dimensional data.

To our knowledge there is no other large scale EHR dataset that is derived from NLP. Related resources in the UK include Hospital Episode Statistics (HES), Clinical Practice Research Datalink (CPRD) and CALIBER. These sources all derive from structured data, primarily diagnosis codes assigned during a primary or secondary care episode. These codes have the advantage that they are manually assigned by trained experts, but the disadvantage is that they are collected primarily for billing/commissioning 'business' purposes and are not necessarily intended to capture all known comorbidities, procedures or medications for the patient. Structured codes are also not error-free [25,26]. The free text narrative, in contrast, is typically much more expressive and detailed as it is designed to create a full record for other clinical staff to rely on (including clinical coding teams).

The challenges of secondary use of real-world data, particularly text data, are not only technical [27]; ethical and legal processes must also be in place. King's College Hospital operates a patient expert-led oversight committee, similar to the model in place at pioneering sites such as the South London and Maudsley CRIS system. Performant NLP is a necessary step to unlocking the research potential of EHRs, but it is not sufficient without similar supporting ethical and legal infrastructure.

## Limitations of this dataset

The great majority of the data in our study is derived from NLP. The NLP model used in this study is well validated on our data and gives good performance, however at scale even a low rate of errors will accumulate. Currently the best approach to improve performance is to manually collect more labelled training data, which is a major bottleneck in clinical NLP [28].

Additionally, we do not know that the NLP performance is stable over time. It is theoretically possible (and likely) that errors may arise due to behavioural differences in the use of vocabulary, especially with changing clinician vocabulary (e.g. splitting of concepts into child-concepts) as well as sociological changes in terminology (e.g. the outdated IDDM/NIDDM, or deprecation of terms like pseudoseizures and increasing use of the term functional neurological disorder). This would only be detectable through examining the corresponding tokens to verify that vocabulary has not changed over time within the dataset. Additionally, care must be taken to curate and validate phenotype definitions for this dataset. We present prevalence estimates for individual SNOMED codes which do not capture all patients with a condition even when the code name is generic (e.g. Heart Failure). To address this we also show prevalence estimates based on the definitions used in the QoF data. Further, future work could potentially use the existing, manually assigned diagnosis codes as a comparison to, or validation of, the NLP assigned codes. Clinical coding data was not available for this study.

Secondly, this dataset should not be directly compared with existing primary care datasets, and national mixed-economy datasets, as these other datasets access different aspects of the healthcare ecosystem (e.g. CALIBER and OpenSafely with primary care). It is possible theoretically to apply the same NLP-process to primary care data sources, mental health data sources or other community records to generate a similar pattern. As this dataset is focused on a single large acute NHS Trust in the UK, it is more analogous to a single site of NHS Digital's HES dataset. It should be viewed as a supplementary NLP-derived mirror of the classical HES dataset.

Our analysis covers all patients at a south London hospital. The demographics of our study cohort are unlikely to be representative of other specific geographic regions and may not generalise to the wider population. It is important that future work investigates whether the patterns identified here are consistent across demographic factors such as ethnic background.

Finally, our analysis is based on SNOMED codes due to their clinical expressiveness. SNOMED is a clinical terminology and not a classification, meaning each code can have multiple parents (a "poly-hierarchy") and as such we are limited to the analysis of individual codes or manually-curated sets of codes (phenotypes). In contrast, classifications such as ICD10 have the advantage that each code belongs in exactly one group, therefore enabling very convenient statistical analysis and aggregation of codes, at the expense of expressiveness. The use of a terminology such as SNOMED imposes certain limitations in large scale analysis where it is infeasible to develop so many specific phenotypes and ensure they do not overlap. Furthermore, the development of clinical phenotype definitions is challenging and requires thorough validation as the choice of codes can significantly affect the derived population [29]. To address this we carried out an analysis of a number of major diseases for which we did develop code sets, and also analysed diseases defined by NHS Digital business rules for the QoF report. To address this limitation at scale it will be necessary to develop either large scale phenotype libraries (such as work by OpenSafely [30] and the HDRUK Phenotype Library [31]) or automated mapping rules to simplify the SNOMED ontology to a structure suitable for statistical analysis [32].

## Conclusion

Large-scale hospital-wide NLP concept capture could provide structured data independently of the structured data entry route of the electronic health record, and support large-scale analysis of multi-morbidity even on legacy datasets. This provides significant scope for hypothesis-generation. It also opens the door towards analyses of 'similar patients' for purposes of virtual trials of multimorbidity. The advent of reliable and scalable NLP methods integrated within

the healthcare data lifecycle [33] brings much more expressive clinical data into a research-ready state and enables all downstream data-dependent activity including population health, business intelligence, and epidemiological and clinical research on real-world data.

## Methods

### Ethical approval

This project operated under London South East Research Ethics Committee approval (reference 18/LO/2048) granted to the King's Electronic Records Research Interface (KERRI). The study operated within the UK legal framework with a research opt-out therefore individual consent was not sought from participants as approved by HRA London South East Research Ethics Committee approval (reference 18/LO/2048). This study was approved by the KERRI committee at King's College Hospital for purposes of evaluating NLP for Clinical Coding (approved Feb 2020) with continued patient oversight.

### Dataset

King's College Hospital has deployed the CogStack platform which enables near-real-time access to data [34], specifically textual data from clinician noting [35], processed using the MedCAT NLP library [36]. Data and access are described online in [37,38]. For this dataset we limited the date range to 2011-01-01 to 2019-12-31 to reduce era-related effects before 2011 and in 2020. The start date captures the availability of data in electronic format, and the end date excludes the impact of coronavirus on the dataset. For this time period, we extracted all clinical notes and letters for male and female patients aged 18–100 at the time the document was created. In total 4571 patients were excluded due missing age or sex information.

### Natural Language Processing for detection of SNOMED concepts

All text was processed using the MedCAT NLP library [36] version 1.3.0 with model hash 8f0d5f743c42a058. In our MedCAT configuration we enable spell checking, ignore words under 3 characters, upper case limit = 4, linking similarity threshold = 0.3. These parameters are tuned to our local dataset. The annotation and meta annotation models were previously trained on the same dataset. Documents were annotated with SNOMED concepts and meta annotated for negation, temporality and experiencer. We considered patients as positive for a SNOMED code if it was detected in their record at least twice and the meta annotations were positive (not negated), experienced by the patient and the temporality was recent or historical. The threshold of 2+ mentions is a heuristic to reduce false positive noise given the scale of the dataset.

In our text preprocessing, documents such as forms and checklists were excluded, as the NLP pipeline is not designed to handle the necessary context of structured documents. Clinical notes are a single document with multiple dated entries, these were split into separate documents. Document spell checking was performed automatically by MedCAT. In total we annotated 212M documents.

### Prevalence

Prevalence estimates were calculated for all 16187 SNOMED disorder codes detected in at least one patient.

### Patient embedding and clustering

We sampled 100,000 patients at random and represented each patient as a vector of the counts for 872 SNOMED disorder codes present in at least 1000 patients in the sample. Vectors were

normalised at patient level to a proportion of their total annotations. This initial representation was reduced to 50 dimensions using PCA, and then further to 2 dimensions using t-SNE with perplexity of 200 and run for 1000 iterations.

We selected 50 clusters based on analysis of cluster inertia with increasing numbers of clusters in mini-batch k-means clustering. The mini-batch k-means algorithm was selected for its scalability. We found that in practice the clusters were qualitatively improved by using agglomerative clustering with ward linkage and a connectivity graph of 100 neighbours. Cluster optimisation was not informed by the subsequent enrichment analysis. The clustering pipeline is available in github [22].

To label clusters with representative diseases, we used the hierarchical structure of the SNOMED ontology. For every patient within a cluster we incremented the count of their detected concepts and all parent terms in the ontology. In this way we can calculate a within-cluster prevalence for all SNOMED concepts, including more general parent concepts. Uninformative concepts such as "Finding" and "Disorder" were excluded and remaining concepts reviewed manually.

### Comparison to national data—UK Quality and Outcomes Framework

The 2018/19 QoF data covering 01 April 2018–31 March 2019 was selected as it captures the most recent year in the dataset. The QoF results are available online [24]. We extracted the corresponding date range from our dataset and applied the SNOMED definitions specified in the NHS business rules version 41 (as used in the 18/19 QoF) available online [23]. Comparisons were made for all disorders defined as SNOMED codes and for a comparable age range. Given the reduced time frame we considered patients with any number of positive mentions in the time frame as positive (i.e. did not apply the threshold of 2 mentions used in all other analysis).

### Statistical comparisons

Continuous variables were compared with a two-sided t-test. Differences in proportions were compared with a two-sided fisher's exact test. All comparisons were corrected for multiple testing using the Bonferroni method.

### Code and software versions

All analysis was carried out in Python 3.8.10, scipy 1.7.1, numpy 1.21.0, and pandas 1.4.2.

### Acknowledgments

This work uses data provided by patients and collected by the NHS as part of their care and support. We would like to thank the patients on the Kings Electronic Records Research Interface (KERRI), the NIHR Applied Research Centre South London, the NIHR Maudsley Biomedical Research Centre, the London AI Centre for Value-based Healthcare, the NHS AI Lab and Health Data Research (UK). The views expressed are those of the author(s) and not necessarily those of the NHS, the NIHR or the Department of Health and Social Care.

### Author Contributions

**Conceptualization:** Daniel M. Bean, Zeljko Kraljevic, Richard J. B. Dobson.

**Data curation:** Daniel M. Bean, Zeljko Kraljevic, Anthony Shek.

**Software:** Daniel M. Bean, Zeljko Kraljevic.

**Visualization:** Daniel M. Bean.

**Writing – original draft:** Daniel M. Bean.

**Writing – review & editing:** Daniel M. Bean, Zeljko Kraljevic, James Teo, Richard J. B. Dobson.

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
