## [Decision Letter · Decision Letter 0]

27 Dec 2022

PDIG-D-22-00287

Hospital-wide Natural Language Processing summarising the health data of 1 million patients

PLOS Digital Health

Dear Dr. Bean,

Thank you for submitting your manuscript to PLOS Digital Health. After careful consideration, we feel that it has merit but does not fully meet PLOS Digital Health's publication criteria as it currently stands. Therefore, we invite you to submit a revised version of the manuscript that addresses the points raised during the review process.

Please submit your revised manuscript within 30 days Jan 26 2023 11:59PM. If you will need more time than this to complete your revisions, please reply to this message or contact the journal office at digitalhealth@plos.org. Please include the following items when submitting your revised manuscript:

We look forward to receiving your revised manuscript.

Kind regards,

Imon Banerjee

Section Editor

PLOS Digital Health

Journal Requirements:

2. Please send a completed 'Competing Interests' statement, including any COIs declared by your co-authors. If you have no competing interests to declare, please state "The authors have declared that no competing interests exist". Otherwise please declare all competing interests beginning with the statement "I have read the journal's policy and the authors of this manuscript have the following competing interests:"

3. Thank you for including an Ethics Statement for your study. Please include a statement that formal consent was obtained (must state whether verbal/written) OR the reason consent was not obtained (e.g. anonymity). NOTE: If child participants, the statement must declare that formal consent was obtained from the parent/guardian.

4. Please provide your detailed Financial Disclosure statement. This is published with the article. It must therefore be completed in full sentences and contain the exact wording you wish to be published.

1. Please clarify all sources of funding (financial or material support) for your study. List the grants (with grant number) or organizations (with url) that supported your study, including funding received from your institution. 

2. State the initials, alongside each funding source, of each author to receive each grant.

3. State what role the funders took in the study. If the funders had no role in your study, please state: “The funders had no role in study design, data collection and analysis, decision to publish, or preparation of the manuscript.”

4. If any authors received a salary from any of your funders, please state which authors and which funders.

5. We ask that a manuscript source file is provided at Revision. Please upload your manuscript file as a .doc, .docx, .rtf or .tex.

Additional Editor Comments (if provided):

While overall recommendation is for a minor revision, please note that reviewer 1 has raised serious questions regarding reproducibility of this work, clustering algorithms performance, data sampling, etc. Please ensure that you can sufficiently answer these questions in the next round of review. Please also improve presentation and readability of the manuscript in light of comments from both reviewers.

Reviewers' comments:

Reviewer's Responses to Questions

**Comments to the Author**

1. Does this manuscript meet PLOS Digital Health’s publication criteria? Is the manuscript technically sound, and do the data support the conclusions? The manuscript must describe methodologically and ethically rigorous research with conclusions that are appropriately drawn based on the data presented.

Reviewer #1: Yes

Reviewer #2: Yes

2. Has the statistical analysis been performed appropriately and rigorously?

Reviewer #1: Yes

Reviewer #2: Yes

3. Have the authors made all data underlying the findings in their manuscript fully available (please refer to the Data Availability Statement at the start of the manuscript PDF file)?

Reviewer #1: Yes

Reviewer #2: Yes

4. Is the manuscript presented in an intelligible fashion and written in standard English?

Reviewer #1: Yes

Reviewer #2: Yes

5. Review Comments to the Author

Reviewer #1: The submitted article is well organized and written. The topic is quite relevant and demonstrates how health-care institutions can leverage their unstructured data to extract knowledge that can help stakeholders to make crucial decisions. This article is also relevant by the number of patients/medical notes included in this work (1.07 M patients).

The main comments that I have for authors are:

- You were able to summarize data and extract valuable information and made a good analysis grouping by gender. Also, you stated very well your limitations. However, you should add a comment regarding ethnicities and disparities. For instance, can it be generalized to all ethnicities and minorities in your dataset? 

- Your sample used for clustering how well describes your population. Does demographics are distributed equally compared to the full dataset? Any biased identified during selection?

- From my opinion and as Data Scientist, its biggest pain point is that I or anyone will not be able to reproduce your work in other datasets. You are not allowed to share dataset according to your Data Availability Statement. Nonetheless, you should be able to share your code for guidance. This would help enormously the scientific community for future work downstream. You can also share the framework that you used for reaching these results (e.g., training pipelines, pre-processing pipelines, etc.)

- I hoped to see performance metrics for your clustering algorithms. For example, which metrics did you used to select the number of clusters?

- Another important aspect that could bring more impact to scientific community is sharing advices and suggestions when you are working with medical notes that are not written in English. Do the tools you used for this work support multilingual inputs?

- Why did not you used python libraries widely used for NLP tasks? (e.g., Spacy, NLTK, etc.). The MedCAT NLP library performs better?

- How did you pre-process your free text? You did not provide enough detail for future reproduction in other datasets.

Reviewer #2: - The URL in the second paragraph of the Results section in page 3 could be set as a citation in the end of the document

- Figure 2 is hard to read and the authors would benefit by deleting "(disorder)" from the labels in the Z axis

- Following the reference to Table 2 in page 5 the authors could add "presented in the appendix of this paper" 

- The disease acronyms provided in the caption of Figure 3 could be provided earlier in the body of the text, as they are mentioned before the Figure is exposed.

- In the paragraph following Figure 3, please specify which clustering algorithm was used and why 

- In the paragraph following Figure 3, please substitute the URL by a standard citation described in the end of the paper. Similarly to the URL appearing in the caption of Figure 4

- I would like to have read the contribution of the paper earlier in the paper (at least summarised in the Introduction section), from the perspective of the last two paragraphs of page 9 and its relation with existing work

6. PLOS authors have the option to publish the peer review history of their article (what does this mean?). If published, this will include your full peer review and any attached files.

**Do you want your identity to be public for this peer review?** For information about this choice, including consent withdrawal, please see our Privacy Policy.

Reviewer #1: No

Reviewer #2: No

---

## [Decision Letter · Decision Letter 1]

16 Feb 2023

Hospital-wide Natural Language Processing summarising the health data of 1 million patients

PDIG-D-22-00287R1

Dear Dr Bean,

We are pleased to inform you that your manuscript 'Hospital-wide Natural Language Processing summarising the health data of 1 million patients' has been provisionally accepted for publication in PLOS Digital Health.

Best regards,

Amara Tariq

Guest Editor

PLOS Digital Health

Thank you for answering reviewers' concerns. Please add the reference mentioned by reviewer 1 in the final version of the manuscript.

Reviewer Comments (if any, and for reference):

Reviewer's Responses to Questions

**Comments to the Author**

1. If the authors have adequately addressed your comments raised in a previous round of review and you feel that this manuscript is now acceptable for publication, you may indicate that here to bypass the “Comments to the Author” section, enter your conflict of interest statement in the “Confidential to Editor” section, and submit your "Accept" recommendation.

Reviewer #1: All comments have been addressed

Reviewer #2: All comments have been addressed

2. Does this manuscript meet PLOS Digital Health’s publication criteria? Is the manuscript technically sound, and do the data support the conclusions? The manuscript must describe methodologically and ethically rigorous research with conclusions that are appropriately drawn based on the data presented.

Reviewer #1: Yes

Reviewer #2: Yes

3. Has the statistical analysis been performed appropriately and rigorously?

Reviewer #1: Yes

Reviewer #2: Yes

4. Have the authors made all data underlying the findings in their manuscript fully available (please refer to the Data Availability Statement at the start of the manuscript PDF file)?

Reviewer #1: Yes

Reviewer #2: Yes

5. Is the manuscript presented in an intelligible fashion and written in standard English?

Reviewer #1: Yes

Reviewer #2: Yes

6. Review Comments to the Author

Reviewer #1: Thank you for addressing all the questions sent before.

Sharing your code helps enormously to the scientific community for future studies.

I have no more concerns to highlight. I just would add the pypi.org link for MedCAT library to the references.

Reviewer #2: I have no further comments.

7. PLOS authors have the option to publish the peer review history of their article (what does this mean?). If published, this will include your full peer review and any attached files.

**Do you want your identity to be public for this peer review?** For information about this choice, including consent withdrawal, please see our Privacy Policy.

Reviewer #1: No

Reviewer #2: No
